# Topical Immunotherapy for Actinic Keratosis and Field Cancerization

**DOI:** 10.3390/cancers16061133

**Published:** 2024-03-12

**Authors:** Laura Bernal Masferrer, Tamara Gracia Cazaña, Isabel Bernad Alonso, Marcial Álvarez-Salafranca, Manuel Almenara Blasco, María Gallego Rentero, Ángeles Juarranz de la Fuente, Yolanda Gilaberte

**Affiliations:** 1Service of Dermatology, Miguel Servet University Hospital, 50009 Zaragoza, Spain; tgraciac@salud.aragon.es (T.G.C.); ibernad@salud.aragon.es (I.B.A.); malvarezs@salud.aragon.es (M.Á.-S.); malmenara@salud.aragon.es (M.A.B.); ygilaberte@salud.aragon.es (Y.G.); 2Department of Biology, Universidad Autónoma de Madrid, 28049 Madrid, Spain; maria.gallego@uam.es (M.G.R.); angeles.juarranz@uam.es (Á.J.d.l.F.)

**Keywords:** actinic keratosis, cancerization field, immunotherapy, immunosuppression, diclofenac disodium, imiquimod, photodynamic therapy, 5-fluorouracil, vitamin D, nicotinamide, solid organ transplant, anti PDL-1

## Abstract

**Simple Summary:**

The primary focus of this review revolves around actinic keratoses (AKs), common skin conditions primarily induced by prolonged sun exposure. These lesions are of particular concern due to their potential progression into skin cancer, thereby warranting in-depth investigation for effective treatment and prevention strategies. There are different approaches to treat AK; some of them target the dysplastic cells, whereas others act on the immunological response of the host to fight against abnormal cells. This review summarizes all the therapies used for AK and field cancerization, whose mechanism of action include the participation of the immune system. Moreover, the research extends to immunosuppressed individuals, mainly organ transplant recipients, who require tailored approaches due to their immune profiles. In essence, this research tries to update the therapeutic approaches for AKs, ultimately lowering the risk of malignant transformation and contributing to improved clinical management.

**Abstract:**

This comprehensive review delves into various immunotherapeutic approaches for the management of actinic keratoses (AKs), precancerous skin lesions associated with UV exposure. Although there are treatments whose main mechanism of action is immune modulation, such as imiquimod or diclofenac, other treatments, apart from their main effect on dysplastic cells, exert some immunological action, which in the end contributes to their efficacy. While treatments like 5-fluorouracil, imiquimod, photodynamic therapy, and nicotinamide are promising in the management of AKs, especially in immunocompetent individuals, their efficacy is somewhat reduced in solid organ transplant recipients due to immunosuppression. The analysis extends to optimal combination, focusing on cryoimmunotherapy as the most relevant. New immunotherapies include resimiquimod, ingenol disoxate, N-phosphonacetyl-L-aspartate (PALA), or anti-PD1 that have shown promising results, although more studies are needed in order to standardize their use.

## 1. Introduction

Actinic keratosis (AK) represents one of the most common premalignant dermatologic conditions, affecting around 25% of the adult population, although its prevalence may be even higher according to some study cohorts evaluating immunosuppression grade or ultraviolet exposure (UV) [1]. These lesions result from cumulative sun exposure and UV radiation damage that cause dysregulation in the cell growth, differentiation, inflammation, and immunosuppression [2].

UV radiation profoundly impacts DNA integrity and immune responses, playing a central role in skin carcinogenesis. UVB induces direct DNA damage, leading to mutations like cyclobutane pyrimidine dimers and 6–4 photoproducts, while UVA produces reactive oxygen species (ROS) that can damage membrane lipids, DNA, and proteins. Beyond DNA, UV exposure triggers a cascade of molecular events, including activation of epidermal growth factor receptors and toll-like receptors or attenuation of Ras and Raf that ends with the production of proinflammatory cytokines such as interleukin (IL)-1, tumor necrosis factor (TNF), and IL-6, and fostering a pro-tumorigenic microenvironment [2].

UV-induced immunosuppression manifests through intricate mechanisms involving DNA damage, cytokine dysregulation, and modulation of immune cell function, collectively undermining immune surveillance and promoting tumor growth. Regulatory T cells (Tregs) expressing the marker forkhead box p3 (Foxp3), recognized for their immunosuppressive properties, play a pivotal role in impeding anti-tumor immune responses, thereby facilitating the progression of AK to squamous cell carcinoma (SCC). In this sense, imiquimod plays a very important role as it is an agonist of toll-like receptors, as we will explain later. Additionally, molecules like platelet-activating factor (PAF) and urocanic acid modulate DNA repair and immune responses, contributing to immunosuppressive pathways [2].

Moreover, UV radiation instigates mitochondrial DNA damage, exacerbating oxidative stress and production of ROS and compromising cellular homeostasis. The tumor suppressor protein p53, pivotal in orchestrating DNA repair and apoptosis, undergoes frequent mutation in UV-induced skin cancers, thereby facilitating tumor progression. Beyond p53, alterations in various gene products such as protein kinase C, epidermal growth factor receptor, and matrix metalloproteinases contribute to the multistep process of keratinocyte transformation and SCC development. Understanding the intricate interplay between UV-induced DNA damage and immunological responses provides crucial insights for devising targeted preventive and therapeutic strategies against UV-associated skin carcinogenesis [2].

Clinically, AKs are characterized by the presence of hyperkeratosis and erythema. While the majority of AKs remain benign, they pose a significant risk of malignant transformation, particularly into SCC, a potentially life-threatening skin cancer [3]. Apart from the concern of SCC development, AKs often present in a field of cancerization (FC), where multiple subclinical lesions coexist in a background of sun-damaged skin. This FC demands comprehensive therapeutic strategies, as addressing individual lesions alone is insufficient to halt disease progression [4].

Over the past few decades, a profound shift in the management of skin cancer has occurred, with a growing emphasis on immunotherapy [5]. Local and systemic immunotherapeutic agents have emerged as pivotal tools for cutaneous tumor, and also, many of the treatments addressed toward AK exert part of their effects modulating the immune system. This paradigm shift is found on a deeper understanding of the immune response within the skin and its pivotal role in regulating tumor development and progression [6]. This immunological effect leverages the body’s natural defense mechanisms to target and eliminate dysplastic keratinocytes and reduce the burden of premalignant and malignant lesions.

In this article, we aim to provide a comprehensive overview of the immunological effect that different treatments for AK and FC have as part of their therapeutic action. We will explore the mechanism of action, efficacy, and safety profiles of various agents, shedding light on their potential to not only treat individual AK lesions but also to modulate the underlying FC effect, which is often overlooked in conventional therapies.

For a better understanding of all the drugs that we analyze in this review, we have decided to divide them according to their mechanism of action in relation to immunogenicity, finding drugs whose main effect is immunomodulation since they act through the immune system to destroy tumor cells. Secondary immunomodulatory drugs are those whose main mechanism is not through the immune system but that consequently have some immunological role and other drugs whose mechanism of action may have involvement of the immune system but this is unknown or under research.

## 2. Material and Methods

All the treatments available for AK and FC have been considered including the following: diclofenac disodium, imiquimod, photodynamic therapy (PDT), 5-fluorouracil (5-FU), anti PDL-1, nicotinamide, vitamin D, as well as future therapies. The bibliographic search for this narrative review was performed at Medline, Embase, and Cochrane databases from January 2002 to July 2023, prioritizing the most recent meta-analyses, systematic reviews, and narrative reviews, using the following keywords as search criteria: actinic keratosis, cancerization field, immunotherapy, immunosuppression, diclofenac disodium, imiquimod, photodynamic therapy, 5-fluorouracil, vitamin D, nicotinamide, solid organ transplant, and anti PDL-1 (Table 1).

## 3. Current Therapies for Actinic Keratosis and Field Cancerization with Primary Immunomodulatory Effect

### 3.1. Imiquimod

Imiquimod, [1-(2-methylpropyl)-1,H-imidazo(4,5-c)quinoline-4-amine], is an immunomodulatory drug approved for the treatment of genital warts by the FDA in 1999 and later approved for the treatment of AK and superficial basal cell carcinoma (BCC) [7].

Being a synthetic activator of toll-like receptors (TLRs), imiquimod has the capability to attach itself to TLR7 and, to a lesser extent, TLR8 on cells responsible for presenting antigens. This interaction initiates the activation of the nuclear factor kappa-B (NF-κB) through the myeloid differentiation factor 88 (MyD88)-dependent pathway. Consequently, this process promotes the maturation of target cells, leading to elevated levels of various proinflammatory cytokines, such as tumor necrosis factor (TNF)-α, interferon (IFN)-α, interleukin (IL)-6, IL-8, and IL-12, along with chemokines like CCL2, CCL3, and CCL4 [8,9] (Table 2).

On one hand, these cytokines amplify innate immunity, and on the other hand, they induce the conversion of T cells into a T helper 1 (Th1) phenotype. This transformation facilitates cell-mediated immune responses by triggering the secretion of IFN-γ from naïve T cells [10] (Figure 1).

Moreover, imiquimod has the capacity to bolster acquired immunity by activating plasmacytoid dendritic cells (pDCs), a distinct subgroup of immune cells expressing high levels of TLR7/9. These cells are responsible for generating type I IFNs, specifically IFN-α and IFN-β. The interaction between imiquimod and TLR7 on pDCs activates the TLR7/MyD88 signaling pathway, resulting in the robust production of type I IFNs. This production is crucial for both innate immune responses and the establishment of the Th1 polarization pattern [9].

Currently available concentrations are 2.5%, 3.75%, and 5%, which are applied three times a week for 12–16 weeks [11]. However, short-term application (4 weeks) has shown similar efficacy to that of long-term application [11].

Regarding the efficacy, a systematic review showed a reduction in AK of 67.5 ± 19.6% at 1 to 3 months, 64.0 ± 13.0% at 3 to 6 months, and 68.0 ± 1.6% at 6 to 12 months after treatment with imiquimod [12]. A network meta-analysis concluded that imiquimod 5% was significantly superior to placebo regarding complete clearance rates (CRs) of lesions after 12-month follow-up (RR, 5.98; 95% CI, 2.26–15.84) [13]. Finally, recurrence rate of imiquimod 5% is 45% (95% CI, 14%-81%), surpassed by cryotherapy and photodynamic therapy with ALA (ALA-PDT) [14].

The most significant drawbacks of imiquimod are its local skin reactions and systemic side effects, both due to its immunological mechanism of action. These include local pain, inflammation, itching, and erythema that are positively correlated with immune activation to systemic flu-like symptoms such as myalgia, fever, and fatigue, mostly due to its role as an interferon inducer [13]. Nevertheless, this treatment is usually well tolerated and only a few patients have to leave it due to adverse effects [11].

Nevertheless, it is crucial to exercise caution when employing imiquimod, particularly in individuals with a personal or family history of autoimmune diseases or inflammatory dermatoses, such as vitiligo, psoriasis, erythema multiforme, and lichen planus. This caution is warranted because imiquimod-induced excessive release of cytokines and the inappropriate suppression of the immune system may contribute to the onset or worsening of these conditions [15].

### 3.2. Diclofenac Disodium

Diclofenac is a well-known non-steroidal anti-inflammatory drug (NSAID) categorized under the phenylacetic acid class, extensively utilized for its anti-inflammatory and analgesic properties. Diclofenac inhibits cyclooxygenase-1 (COX-1) and cyclooxygenase-2 (COX-2), resulting in the inhibition of the arachidonic acid cascade and preventing the formation of thromboxane and prostaglandins (PGs) [11,16]. COX-2 expression and PG production are induced following skin exposure to UV radiation [17]. These events play a key role in inflammatory processes in the skin that are believed to be associated with progression from AK to SCC [18]. Multiple investigations in mouse models have underscored the significance of COX-1, COX-2, prostaglandin E2 (PGE2), and PGE2 receptors in the development and progression of skin tumors [19].

Topically, diclofenac is indicated for the treatment of AK due to its mechanism of action based on the inhibition of angiogenesis and cell proliferation resulting in keratinocyte apoptosis [11]. Nevertheless, recent studies indicate that diclofenac also modulates the immune response, inducing the infiltration of CD8+ T lymphocytes and IFN-γ [20] (Table 3).

Maltusch et al. investigated the modes of action of diclofenac 3% with hyaluronic acid 2.5% (HA). This research observed a significant decrease in the expression of inflammatory markers such as COX-2 (epidermis), CD3, and CD8 compared to the levels before treatment (*p* = 0.006, 0.005, and 0.013, respectively). Additionally, the post-treatment expression of markers associated with apoptosis and/or cell cycle arrest (p53 and p21) showed significant reductions compared to both healthy skin (p53 [*p* = 0.011]) and pre-treatment levels (p21 [*p* = 0.003]). The post-treatment expression of CD31, a marker of angiogenesis, was also notably lower compared to pre-treatment levels (*p* = 0.015). They concluded that the clinical improvements in AKs associated with diclofenac 3%/HA 2.5% primarily result from its anti-inflammatory and anti-angiogenic effects, with contributions of its effect on proliferation and apoptosis [21].

Diclofenac is formulated at 3% in hyaluronic acid at 2.5% and applied two times a day for 60–90 days [11,22]. The percentage of AK clearance after 1 to 3 months post-treatment is 36.9 ± 9.5% [13] and after 6 months is 45% [13], and is one of the less-effective treatment options. In addition, according to a systematic analysis, diclofenac had the highest recurrence rates compared to the rest of treatments analyzed [14].

Its side effects include irritation, itching, stinging, edema, pain, and inflammation similar to other topical treatments for AK [22]. However, their intensity and incidence seem to be lower than that for 5-fluorouracil or imiquimod [16].

## 4. Current Therapies for Actinic Keratosis and Field Cancerization with Secondary Immunomodulatory Effect

### 4.1. Photodynamic Therapy

Photodynamic therapy (PDT) is a minimally invasive therapeutic modality in which a compound with photosensitizing properties selectively accumulates in tumor cells [23]. The subsequent activation of the photosensitizer (PS) with a light source with the appropriate wavelength generates reactive oxygen species (ROS), mainly singlet oxygen, which is responsible for the cytotoxic effect of neoplastic cells and tumor regression [23,24]. 

PDT can induce cellular death by different pathways, depending on the amount, localization, and type of PS, light dose, oxygen level, and cell type. Therefore, PDT can induce cellular death through apoptosis, necrosis, and autophagy [25,26]. PDT can also trigger an immune response by releasing damage-associated molecular patterns (DAMPs) from the dying cells [27,28]. DAMPs are molecules that signal tissue damage and activate innate and adaptive immunity. Some examples of DAMPs are calreticulin (CRT), ATP, and HMGB1 (high mobility group box 1 protein) [27,28]. These molecules can bind to specific receptors on antigen-presenting cells (APCs) and stimulate their maturation and migration to lymph nodes [25,27,28]. There, they present tumor antigens to T cells and initiate an anti-tumor immune response. PDT can also modulate the tumor microenvironment by affecting cancer-associated fibroblasts (CAFs) [27]. CAFs are fibroblasts that support tumor growth, invasion, and angiogenesis. PDT can reduce the number and activity of CAFs, thus inhibiting their pro-tumoral effects [27] (Figure 1).

The compounds commercialized in Spain to be used in PDT for AK are methyl-aminolevulinate cream (MAL) and 5-aminolevulinic acid nanosomized in gel (ALA). They both are precursors of intracellular PS that accumulates inside the cells, the protoporphyrin IX (PpIX), a PS of heme group synthesis. Regardless the PS used, pre-treatment is generally recommended to remove crusts, hyperkeratosis, and superficial scaling, facilitating the photosensitizer penetration and improving effectiveness [29,30]. Curettage is a common and efficient pre-treatment method. However, there are other methods that can be effective to eliminate hyperkeratosis such as the Erb:YAG or CO2 laser, which in some studies have shown superiority over curettage [31].

The efficacy of conventional PDT (cPDT) for AK and non-hyperkeratotic FC is high, with cure rates around 80–90% [32]. The technique has become essential in managing patients with multiple AK and FC and is especially recommended for immunosuppressed transplant recipients [29,30,32]. A comparative study between MAL and ALA suggests slightly higher efficacy with ALA, but with a significantly more pronounced local reaction [33].

One drawback of conventional PDT is the pain experienced during illumination, leading to the development of daylight PDT, which has been shown to be no inferior in efficacy compared to the conventional form but better tolerated [34,35]. Overall, PDT remains a valuable tool, particularly in managing AK and FC, with a high degree of recommendation and evidence in guidelines [29].

### 4.2. 5-Fluorouracil

5-fluorouracil (5-FU) is an anticancer agent from the antimetabolite group and a pyrimidine analogue widely used as a chemotherapeutic. Its mechanism of action is based on the inhibition of thymidylate synthetase [7,16], an enzyme necessary for DNA synthesis that prevents cell proliferation and produces cell apoptosis [11,36]. In addition, it can increase the expression of p53, a mechanism by which it would enhance the apoptosis of dysplastic cells, making it useful in the treatment of the FC [22]. 

The immune response triggered by 5-FU chemotherapy can be broadly categorized into two main aspects: (1) the reduction or elimination of immunosuppressive myeloid-derived suppressor cells (MDSCs) and (2) the induction of immunogenic cell death through the activation of DAMPs. The primary immune cell types contributing to immunosuppression and immune evasion include MDSCs, tumor-associated macrophages (TAMs), type-II natural killer T cells, and regulatory T cells (Tregs) [37,38]. 

MDSCs, including monocytic MDSCs and granulocytic polymorphonuclear (PMN), represent a heterogeneous population of immature myeloid cells that fail to undergo full differentiation into monocytes and neutrophils. These cells suppress both innate and adaptive immune responses. In healthy individuals, MDSCs exist in low numbers and play a role in routine immune surveillance and tissue repair processes. However, during infection, inflammation, or cancer, MDSCs rapidly expand and accumulate in the blood, bone marrow, peripheral lymphoid organs, and tumors, as demonstrated in murine models and observed in humans [37,39].

The concentrations of 5-FU currently available are 0.5%, 1%, 4%, and 5% [22]. 5-FU at 5% applied 2 times daily for 2–4 weeks has been extensively studied as field treatment in immunocompetent as well as immunocompromised patients. Jansen et al. [12] analyzed the cumulative probability of treatment success for 5-FU at 12 months, which was 74.7% (95% confidence interval [CI], 66.8 to 81.0), much higher than that of the other treatments analyzed. Worley et al. [13] in a systematic review of treatments for AK found a reduction of 80.1% of lesions at 1–3 months and 67.4% at 3–6 months with a recurrence rate of 27% at 12 months with 5-FU at 5%. However, another meta-analysis showed recurrence rates at more than 12 months of 52% (95% CI, 38–66%) [40]. Krawtchenko et al. [41] analyzed the clinical and histological efficacy of 5-FU 5% vs. other treatments and observed a clinical cure in 96% of patients (23 of 24) vs. a histological cure of 67% (16 of 24) after the end of the treatment. At 12 months, 57% of the patients (13 of 23) did not present clinical recurrence of the cleared lesions. 

The adverse effects of 5-FU are mainly scaling, erythema, crusting, stinging, pruritus, and burning [11,16,22,42]; therefore, regimens with lower concentrations have been investigated to diminish them. 5-FU 0.5% applied twice daily for 4–6 weeks appears to have similar efficacy and better tolerability, which could be desirable in elderly patients to decrease systemic absorption of 5-FU and to improve adherence, with total lesion clearance rates of 34% vs. 49% for 5-FU 5% [43]. Also, the 4% concentration, applied once per day for 4 weeks, has shown to significantly reduce adverse effects while maintaining efficacy [11]. A network meta-analysis showed that 5-FU 5% followed by 5-FU 4% seemed to be superior in efficacy than other treatments for AK [44]. Additionally, pretreatment with petrolatum seems to reduce local skin reactions of the 5-FU 5% concentration without affecting the efficacy [11]. 

## 5. Other Therapies Used Off-Label for Actinic Keratosis and Cancerization Field with Immunomodulatory Effect

### 5.1. Vitamin D

Calcitriol and its analogues are widely used in psoriasis. When applied topically to epithelial cells, calcitriol and other analogues induce the expression of thymic stromal lymphopoietin (TSLP), a cytokine structurally similar to IL-7. The receptor complex formed by TSLP and IL-7 on CD11c + dendritic cells lead to the production of Th2-enriched inflammation by inducing chemokines like CCL17 and CCL22. Additionally, TSLP inhibits Th1 and Th17 differentiation, suggesting a role in reducing psoriatic plaques. Studies reveal that vitamin D analogues stimulate TSLP and cathelicidin, suppressing inflammatory cytokines and improving psoriatic plaques.

TSLP’s immunomodulatory roles extend to inflammation and cancer. While promoting neoplastic growth in breast and pancreatic tissues, TSLP exhibits a protective role against skin neoplasm carcinogenesis. TSLP directly signals CD4+ and CD8+ T-cells in the skin, inhibiting cancer development, particularly β-catenin-dependent skin tumors. However, other studies have also shown that TSLP secreted by keratinocytes is a potent inductor of robust antitumor immunity in the barrier-defective skin. Based on these findings, Demehri et al. have carried out several in vivo studies and clinical trials using calcipotriol (a low-calcemic vitamin D analogue) as an inductor of TSLP alone and in combination with 5-FU. Their results came up as promising in the field of prevention, as they showed that this combination was able to decrease the risk of SCC development up to three years after the treatment of AK. AK biopsies after this combination exhibited increased TSLP expression and massive CD4+ T cell infiltrates that persisted in the epidermal niche, being responsible for this preventive outcome [45,46,47,48]. Additionally, topical calcitriol and calcipotriol have been used previous to ALA or MAL PDT, enhancing the efficacy but also the side effects [49,50].

### 5.2. Nicotinamide

Nicotinamide (pyridine-3-carboxamide) (NAM) is a water-soluble amide active form of vitamin B3 or niacin (pyridine-3-carboxylic acid) [51]. NAM and niacin are precursors for the synthesis of nicotinamide adenine dinucleotide (NAD^+^) and the phosphorylated derivative NADP^+^. NAD^+^ is an essential co-enzyme of redox reactions for adenosine triphosphate (ATP) production and for several other metabolic processes. Beyond its role as a co-enzyme, NAD^+^ may act as a substrate for specific NAD^+^-consuming enzymes and influences cellular responses to genomic damage through different mechanisms [51].

Interestingly, UVB-irradiated keratinocytes have shown a marked decrease in NAD^+^, which could increase the tumorigenic potential secondary to a loss of adequate energy production required for DNA repair [52]. Nicotinamide appears to exert its UV-protective effects on the skin via its role in cellular energy pathways. NAM prevents UV-induced cellular ATP loss and protects against UV-induced glycolytic blockade, by increasing DNA repair when keratinocytes are supplemented with NAD+ after UV irradiation and also reduces UV-induced suppression of immunity and inflammation [52] (Figure 2). 

In 2013, Monfrecola et al. conducted a study on the impact of NAM on UV-induced inflammatory cytokine production in human keratinocytes, which was conducted ex vivo. HaCaT cells, an immortalized human keratinocyte cell line, were exposed to NAM treatment and subsequently subjected to UV irradiation. HaCaT cells, which received prior treatment with NAM, exhibited a noteworthy decrease in the expression of IL-6, IL-10, MCP-1, and TNF-α mRNA, suggesting the efficacy of NAM in inhibiting local inflammatory responses [53].

All of these actions contribute to restrict the development of a cancer-promoting environment and enhance the drug’s capacity as a promising preventive treatment [54]. 

Most studies using NAM have been performed by oral administration. In Phase II and III trials, nicotinamide at a dosage of 500 mg twice daily has proven to be an effective and well-tolerated preventive approach against AK lesions when employed for one year [51].

A phase II dose-optimizing trial in 2012 determined that twice daily supplementation with NAM 500 mg was effective for reducing AK counts with minimal adverse drug reaction compared to daily dosing [51]. In a meta-analysis conducted by Mainville et al. [55] in 2022, they concluded that the low cost and over-the-counter accessibility of NAM support its relevance in the tertiary prevention of skin cancers in the immunocompetent population. However, these positive results have not been demonstrated in solid organ transplant recipients; in a 12-month placebo-controlled trial with a total of 158 enrolled participants, oral nicotinamide therapy did not significantly decrease the number of keratinocyte cancers or AK [56].

To the best of our knowledge, there are only a few studies available regarding the topical use of NAM for AK. Sivapirabu G et al. [57] designed a randomized controlled trial, which involved healthy volunteers who were exposed to different types of UV radiation, applying NAM after each exposure. The results showed that the 5% concentration especially counteracts the effect of UV radiation enhancing the cellular energy metabolism and p53 activity. NAM also effectively protects against the immunosuppression caused by UVB, UVA, and repeated UV exposures. This suggests that NAM holds promise as a preventive agent against skin cancer, including the highly immune-suppressive longwave UVA radiation.

Another trial compared topical NAM 1% twice a day vs. placebo to treat forty-one immune-competent adults with a minimum of four nonhyperkeratotic AKs in various treatment areas [58]. After 3 months, the group treated with NAM showed a higher significant reduction in AK (21.8% reduction) compared to the vehicle (10% reduction); however, this difference was not statistically significant at 6 months. Site-specific differences were not observed, but there was a more pronounced reduction in AKs in men compared to the vehicle at both 3 and 6 months.

Finally, applying sunscreen daily along with topical NAM can lead to a reduction of approximately 40% in the number of AKs. This implies that the combination of NAM and sunscreen could potentially provide a higher level of immune protection compared to using nicotinamide or sunscreen alone.

## 6. Topical Immunotherapy Associated with Cryotherapy: Cryoimmunotherapy

Cryoimmunotherapy is increasingly used in dermatology. Cryoimmunotherapy is a medical approach that combines cryotherapy with immunotherapy, which enhances the body’s immune response to target skin cancers and also AK. It has been hypothesized that cryotherapy could lead to tumor reduction and the subsequent release of tumor antigens that would trigger an immune response. These antigens can then be recognized by immune cells, such as macrophages and dendritic cells, which tend to become active in the presence of imiquimod. This process might support the development of a targeted immune response against AK.

While imiquimod itself can activate both innate and adaptive immune cells, leading to some degree of AK lesion reduction by cytolytic CD8+ T cells, the presence of these released antigens enhances its efficacy. This phenomenon also clarifies why imiquimod surpasses cytoreductive therapy alone in terms of effectiveness [59] (Figure 3).

The most commonly used combination in cryoinmunotherapy is with imiquimod and photodynamic therapy. Thus, a recent systematic review and meta-analysis about the safety and efficacy of the combination of cryotherapy and PDT with imiquimod in patients with AK screened a total of 1031 studies and finally included five different studies [60]. Three studies compared the effect of imiquimod/cryotherapy to cryotherapy alone or to imiquimod/placebo. The other two studies compared the effect of imiquimod/PDT to imiquimod alone and to PDT alone [61,62,63,64,65]. The meta-analysis revealed that imiquimod/cryotherapy significantly improved complete clinical clearance in AK patients (OR: 6.26; 95%CI: 1.56–24.1; *p* = 0.01). Importantly, no serious adverse effects were observed across the different treatment options. They concluded that combining PDT or cryotherapy with imiquimod was more effective for complete AK resolution than using imiquimod alone.

However, not only PDT and imiquimod have been combined with cryotherapy, ingenol mebutate, a modulator of eight protein kinase C (PKC) isoenzymes, used before cryotherapy leads to a lower local skin reaction. This probably reduces patient discomfort and improves compliance [66]. The possibility of adding two different mechanisms of action seems to be a good strategy to improve results and overcome the limitations of each drug, reducing the risk of developing resistances.

## 7. Topical Immunotherapy for Actinic Keratosis in Organ Transplant Recipients

OTRs are a special population for usage of treatments with immunological mechanism to treat AK and FC. The treatments used include PDT, imiquimod, diclofenac, and to a lesser extent, 5-fluorouracil. In general, these treatments are less effective in OTRs due to their immunosuppression, resulting in a lower response compared to immunocompetent patients. Additionally, a strong stimulation of the immunological system could alter the immunotolerance to the organ transplant. 

PDT has the strongest evidence to be used in OTR. Dragieva et al. found a CR of 68% for AK at 12 weeks after one or two sessions of cPDT with ALA compared to 89% CR in immunocompetent patients [67]. Usually, the number of PDT sessions are two and the photosensitizer is MAL, reaching a CR rate of 71% or 90% of the lesions at 12 or 16 weeks, respectively [68,69]. Hasson et al. observed CR at 12 and 24 weeks for AK in 10 organ transplant recipients after one session of PDT and in 6 recipients after two sessions [70]. Perret et al. compared two cycles of cPDT-MAL separated by 1 week with the application of 5-FU twice daily for 3 weeks in two different areas of each patient in eight patients. The CR rate sat one and three months were 89% on the cPDT side and 0% and 11%, respectively, on the 5-FU-treated side [71]. 

Some treatments combined with cPDT enhance its efficacy. Togsverd-Bo et al. compared ablative fractional laser (LFA) + daylight-PDT, daylight-PDT, cPDT, and LFA in the same patient. They found a CR rate per lesion of 74%, 46%, 50%, and 5%, respectively, after one treatment session [72]. Jambusaria et al. conducted sequential treatment involving gentle curettage, application of 5-FU at 5% twice daily for 5 days, and cPDT with 1 h incubation on the 6th day in four organ transplant recipients. They achieved complete or near-complete response in all patients at one month and six months of follow-up [73]. In another study, microneedling was performed prior to cPDT-MAL, with three sessions separated by 2 weeks. At 9 months, 83% maintained CR, and 17% experienced recurrences [74].

There are also studies evaluating the efficacy of cPDT and daylight-PDT in preventing the appearance of new AK, showing that fewer new lesions develop with repeated treatments of the FC [75,76,77,78,79,80,81].

Regarding 5-FU, its application twice daily for 3 weeks resulted in clearance rates per lesion of 98% after 2 months and 79% after 12 months of follow-up [82].

Two studies have evaluated the efficacy of imiquimod 5%, three times per week for 16 weeks in a 100 cm^2^ area, reaching a clearance rate of 62% [83,84]. No rejection or functional deterioration of the transplanted organ was reported in these studies; however, one case of acute renal failure has been reported in one kidney transplant patient treated for common warts with imiquimod on an area larger than 100 cm^2^ [85].

In one study, complete remission of AK was observed in 41% of patients after applying diclofenac 3% twice daily for 16 weeks, with a recurrence rate of 55% [83]. In another study with the same treatment regimen, complete remission was achieved in 50% of patients [86].

Based on the studies conducted so far, the highest clearance rates for AK in OTR are associated with MAL-PDT, with increased efficacy achieved through two PDT treatment cycles separated by 1–2 weeks or combination with other treatments such as LFA, 5-FU, or microneedling. Treatment with 5-FU also achieves higher clearance rates than diclofenac and imiquimod, but more studies are needed (Figure 4).

## 8. Future Perspectives in Immunotherapy for Actinic Keratoses

The diagnosis of AK is eminently clinical, but we cannot ignore that sometimes discerning an AK from Bowen’s disease can be complex if there is a lot of hyperkeratosis or the FC is very damaged. For this reason, there is a general interest in finding non-invasive methods to help the diagnosis and management of these injuries. Devices like OCT or LC-OCT have emerged as tools that allow optimizing efficiency and accuracy in the evaluation of actinic keratosis, thanks in turn to the application of artificial intelligence that, through algorithms, allow better categorization of the degree and depth of the lesions, which can help guide treatment and reduce the number of biopsies [87,88].

New drugs are currently being developed with the ambitious goal of increasing clearance rates of AKs, optimizing treatment regimens, and minimizing local side effects [89]. Different mechanisms of action are under investigation, including EGFR/ErbB2 antagonists (sinechatechins), beta-tubulin antagonists (paclitaxel), Na^+^ K^+^ transporting ATPase inhibitors (furosemide and digoxine), or voltage-dependent anion channel (VDAC)/ hexokinase 2 (HK2) modulators (tuvatexib), among others [90]. 

Specifically, resiquimod (S-28463, R-848, VML600) acts as a topical immune-modifier directly on TLR7/8, resulting in the activation of myeloid and plasmacytoid dendritic cells. Moreover, it has been suggested that this agent may improve cancer immunotherapy by reducing the immunosuppressive effect of MDSC [91]. Topical resiquimod induces more IL-12 and it is 10- to 100-times more potent than imiquimod [89,92]. A phase II dose-ranging study found that resiquimod 0.01–0.03% in a dosing regimen of three times per week for 4 weeks covering an area of 25 cm^2^ achieved complete clearance rates of 77.1–90.3% with fewer side effects than higher concentrations (0.06 and 0.1%) [93]. A further multicenteric, partly placebo-controlled, double-blinded clinical trial in 2019 (NCT01583816) showed the following: (1) resiquimod was superior than placebo in all treatment groups; (2) resiquimod 0.03% was more effective than the 0.01% concentration; (3) shorter treatment regimens (7×/2 weeks and 5×/1 week) offer similar effectiveness than longer schedules; and (4) dosing regimens until erosion is achieved could be equally effective and may be used as a personalized approach in the treatment of AKs [94]. Most adverse effects associated with resiquimod occur at the application site and mainly consist of erythema, edema, or scabbing, may be associated with burning, pain, or pruritus, and are concentration-dependent. Moreover, influenza-like symptoms secondary to cytokine release have been described in the higher concentration groups [93,94]. While these results are promising, further data on long-term clearance, large-field application regimens, and direct comparisons with standard therapy, are needed to establish the role of resiquimod in the current armamentarium against AKs [95].

There is currently a growing interest in elucidating the possible role of human papillomaviruses (HPVs) in keratinocyte skin cancer tumorigenesis. In fact, a recent systematic review found that β and γ HPV genera were detected in 58 and 40% of AKs screened for HPV presence, respectively [96]. Although it is well known that HPV can be found in healthy skin, the study conducted by Galati et al. showed that some γ-HPV types are more frequent in AKs compared to normal skin [97]. In this line, a small non-controlled case series with 12 immunocompetent patients conducted by Wenande et al. found that the administration of a nonavalent HPV vaccine (GARDASIL9^®^; Merck & Co., Whitehouse Station, NJ, USA) could be a potentially effective therapeutic approach in combination with the standard treatment for AKs. The authors reported an average of 85% reduction in the total AK burden 12 months after initiation of HPV vaccination [98]. These preliminary results, however, should be confirmed by well-designed, large, randomized, and controlled trials focusing on the pathomechanisms underlying this HPV–AKs relationship and its potential therapeutic role.

Another promising treatment is ingenol disoxate (LEO 43204), a novel ingenol derivative with a dual mechanism of action, involving direct cellular cytotoxicity and activation of PKC, with subsequent induction of a local immune response. This agent shows better chemical stability properties than ingenol mebutate, which facilitates storage of the product at ambient temperatures [99]. Phase II and III trials have shown that ingenol disoxate is superior to the vehicle and well-tolerated as field therapy for AKs located on the scalp, full face, or chest [100,101,102], with favorable cosmetic outcomes and patient satisfaction [103]. However, a slight increase in skin cancer (SCC, BCC, and Bowen’s disease) in the treatment area [102], along with the known association between SCC and ingenol mebutate [104], will probably limit the development of this agent for the treatment of AKs/FC in the future (personal opinion of the authors).

PD-1 receptor is expressed on the cell surface of activated T cells. It binds to the programmed cell death ligand 1 (PDL-1) produced by tumor cells, inhibiting the function of T cells. This interaction can be impeded with PD-1 and PDL-1 inhibitors, facilitating the generation of stronger immune responses against tumor cells. In this way, anti-PD-1 and PDL-1 inhibitors are used as immunotherapy for cancers with very good results [105]. Monoclonal antibodies against PD-1 have been tested in clinical trials for SCC including cemiplimab for the treatment of advanced cutaneous SCC and pembrolizumab for recurrent/metastatic cutaneous SCC [106]. It has been observed that T cells, Langerhans cells, and the expression of PDL-1 gradually increase during the progression from AK to in situ and invasive SCC, justifying the potential efficacy of their treatment with anti-PD-1/PDL1 immunotherapy [107].

In 2022, a case of inflammation of AK was reported in a patient after receiving atezolizumab treatment, an anti-PDL-1 antibody, for diffuse large B-cell lymphoma. The authors suggest that atezolizumab may also be a selective therapy for AKs, and future studies could evaluate the efficacy of topical anti-PDL-1 agents as immunotherapy for AK [105].

Other drugs are in very early stages of development, such as N-phosphonacetyl-L-aspartate (PALA). PALA inhibits pyrimidine nucleotide synthesis and activates pattern recognition receptor nucleotide-binding oligomerization domain 2. Its anti-neoplastic activity has been associated in a mouse model with increased expression of the antimicrobial peptide cathelicidin and increased recruitment of CD8^+^ T cells and F4/80^+^ macrophages to the tumors, demonstrating both immunomodulatory and anti-proliferative effects. These findings postulate PALA as a good candidate as an effective alternative to current standard-of-care NMSC therapies [108].

In the same way that immunotherapy is currently one of the mainstays of cancer therapy innovation, the development of new agents that address this therapeutic pathway for the treatment of AKs as “cutaneous pre-cancer” would be desirable.

## 9. Conclusions

In conclusion, this review underscores the complexities of AK treatment and the potential offered by various immunotherapeutic approaches. Although there are treatments whose main mechanism of action is immune modulation, such as imiquimod or diclofenac, other treatments, apart from their main effect on the dysplastic cells, exert some immunological action, which in the end contributes to their efficacy. While treatments like 5-FU, imiquimod, PDT, and NAM are promising in the management of AKs, especially in immunocompetent individuals, their efficacy is somewhat reduced in solid organ transplant recipients due to immunosuppression.

The future of AK immunotherapy appears promising, with novel agents like resiquimod and ingenol disoxate demonstrating favorable results. The detection of HPV in AKs also raises intriguing possibilities for HPV vaccination in AK management. However, it is essential to balance therapeutic efficacy with safety, considering the increased risk of skin cancer with certain treatments. Overall, the multifaceted strategies discussed in this review offer clinicians a range of tools for AK management, including the possibility of specific treatment combinations, allowing them to personalize the therapy to the patient needs and maximize therapeutic outcomes. Further research and clinical trials will undoubtedly provide more insights into the evolving landscape of AK immunotherapy, with the ultimate goal of reducing the burden of this precancerous condition and its potential progression to skin cancer.

## Figures and Tables

**Figure 1 cancers-16-01133-f001:**
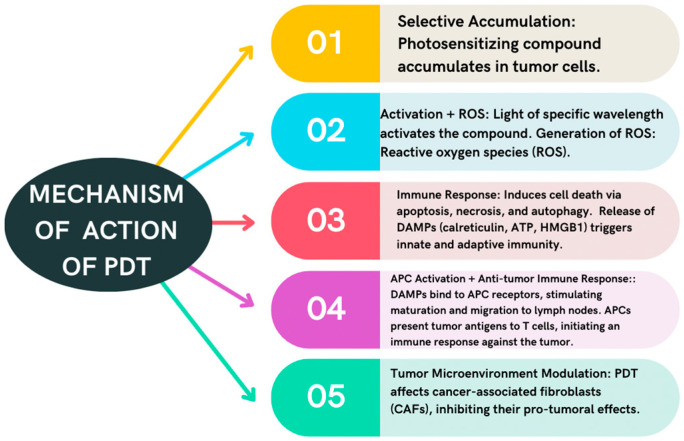
Mechanism of action of PDT. This image summarizes the most important effects of PDT in the tumor cells.

**Figure 2 cancers-16-01133-f002:**
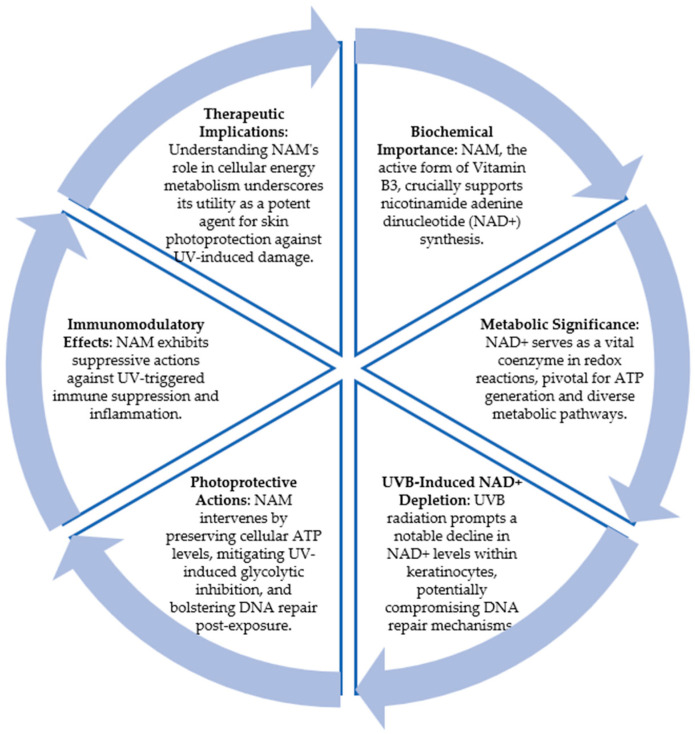
Nicotinamide and skin protection. This figure summarizes the role of nicotinamide in protecting against skin cancer and its mechanism of action.

**Figure 3 cancers-16-01133-f003:**
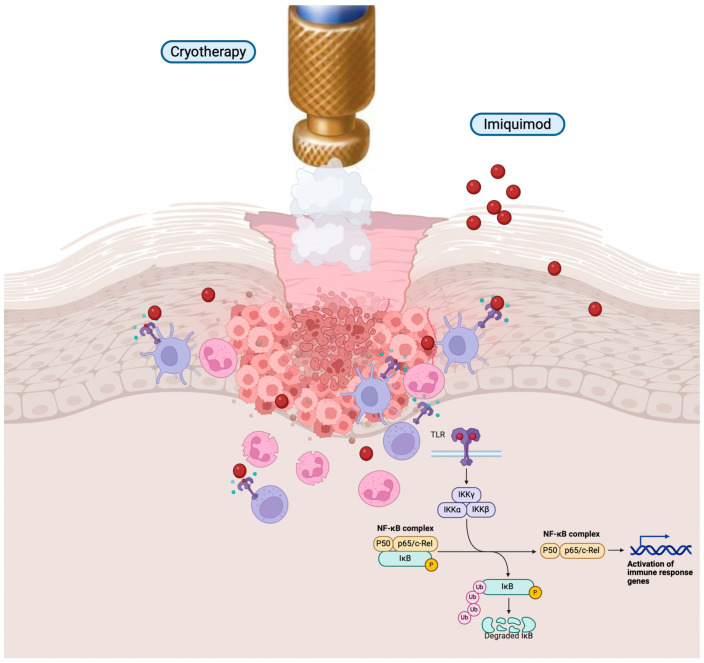
Immunomodulatory mechanisms of cryoimmunotherapy. This figure summarizes all the important keys in how cryoimmunotherapy triggers a molecular cascade that begins with TLRs and the NF-kB complex and results in the activation of the immune system.

**Figure 4 cancers-16-01133-f004:**
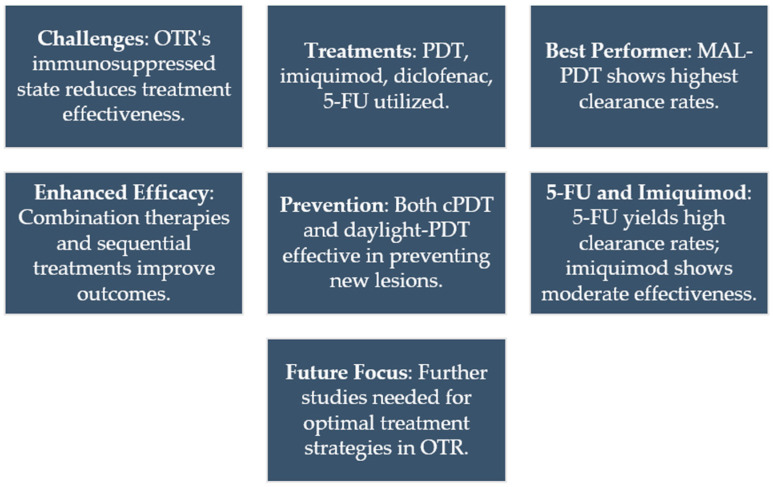
Treatment of AK in TOS. This figure summarizes all the important keys in the treatment of AK in TOS.

**Table 1 cancers-16-01133-t001:** Structure of this review.

Primary Immunomodulatory Effect	Secondary Immunomodulatory Effect	Immunomodulatory Effect under Research	Topical Immunotherapy Associated with Cryotherapy	Organ Transplant Recipients	Future Perspectives
- Imiquimod - Diclofenac disodium	- Photodynamic therapy - 5-fluorouracil	- Vitamin D - Nicotinamide	- Photodynamic therapy - Imiquimod - Ingenol mebutate	- Photodynamic therapy - Imiquimod - 5-fluorouracil - Diclofenac disodium	- Resiquimod - GARDASIL9^®^ - Ingenol disoxate - Anti-PD-1 - N-phosphonacetyl-L-aspartate

This table summarizes the structure followed in this review by dividing the different drugs according to their immunomodulatory mechanism of action.

**Table 2 cancers-16-01133-t002:** Immunomodulatory mechanisms of imiquimod.

FDA Approval	Imiquimod approved for genital warts, AK, and superficial BCC.
TLR Activation	Binds to TLR7 and TLR8, initiating NF-κB pathway via MyD88.
Cytokine Release	Increases proinflammatory cytokines (TNF-α, IFN-α, IL-6, IL-8, IL-12) and chemokines (CCL2, CCL3, CCL4).
Innate Immunity Enhancement	Amplifies innate immunity.
Th1 Phenotype Induction	Promotes T cell conversion to Th1 phenotype, stimulating IFN-γ secretion.
pDC Activation	Stimulates plasmacytoid dendritic cells (pDCs) expressing TLR7/9.
Type I Interferon Production	Triggers robust production of IFN-α and IFN-β.
Immune Response Amplification	Enhances both innate and acquired immune responses. Principio del formulario Final del formulario

This table summarizes the immunomodulatory mechanism of imiquimod.

**Table 3 cancers-16-01133-t003:** Summary of approved topical treatments.

Name of the Treatment	Mechanism of Action	Posology	Adverse Effects
5-fluorouracil	Pyrimidine analogue	5% 2 times daily for 2–4 weeks	Scaling Pruritus
	Inhibition of thymidylate synthetase	4% 1 time daily for 4 weeks	Erythema Burning
	Induces cell apoptosis	0.5% 2 times daily for 4–6 weeks	Crusting
Imiquimod	Activates toll-like receptor 7	2.5%, 3.75%, and 5% three times a week for 12–16 weeks	Local reactions Erythema
	Production of TNF-α, IFN-γ, IFN-α, and IL-12 Induces cell apoptosis		Systemic flu-like symptoms
Diclofenac disodium	Inhibition of COX-1 and COX-2	3% 2 times a day for 60–90 days	Irritation Itching Erythema
	Inhibition of angiogenesis and cell proliferation Induces cell apoptosis		
Photodynamic therapy	Production of ROS and releases DAMPs Modulates CAFs	Daylight: One session curettage of the lesions followed by application of the photosensitizer, 30 min incubation, and sun exposure for 2 h. Conventional: Two sessions curettage of the lesions followed by application of the photosensitizer, 3 h incubation, and subsequent lighting.	Pain Burning Erythema Irritation Pruritus
	Induces cell apoptosis, necrosis, and autophagy		

This table summarizes the most frequently used treatments currently approved for actinic keratoses. In it, we can see the mechanism of action of each of them, dosage, and most frequent adverse effects. It should be noted that the adverse effects overlap between the different therapies.

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
