# Peer review of "Topical Immunotherapy for Actinic Keratosis and Field Cancerization"

_cancers, 2024, doi:10.3390/cancers16061133_

Round 1

Reviewer 1 Report

Comments and Suggestions for Authors

The article titled "Topical Immunotherapy for Actinic Keratosis and Field Cancerization" presented by Bernal Masferrer et al. discusses the state of advancement of immunomodulatory drugs for actinic keratosis presented as premalignant lesions, also in the field of cancerization. The methodology of articles and keywords included to carry out the topic review has been described. Although I consider that the article involves a logical description of the various sections proposed, I find some aspects insufficiently discussed or not addressed that should be included:

  1. The introduction is very brief and only presents the topical disease of the review. A theoretical framework related to immunological aspects and how different types of cells contribute to therapeutic action could be included. This could be a separate section of the review.

  2. I believe Table 1 does not provide relevant information for the article. The description of the sections to be addressed in the review could be included in the material and methodology section.

  3. Describing the difference between primary immunomodulatory and secondary immunomodulatory effects would help understand the differences between the treatments grouped for each of these sections.

  4. An introduction in each section would help understand why the drugs shown are being presented and described.

  5. The review could improve its quality and readability if the authors include images or diagrams either created by them or obtained from other articles to help exemplify and explain various aspects.

Comments on the Quality of English Language
  1. Care should be taken to address grammatical, syntactic, and repetitive errors found throughout the article.

Author Response

Thank you very much for taking the time to review this manuscript. Please find the detailed responses below and the corresponding revisions.

Comments 1 and 3:

  1. The introduction is very brief and only presents the topical disease of the review. A theoretical framework related to immunological aspects and how different types of cells contribute to therapeutic action could be included. This could be a separate section of the review.

  2. Describing the difference between primary immunomodulatory and secondary immunomodulatory effects would help understand the differences between the treatments grouped for each of these sections.

Response 1 and 3: First of all, thank you very much for your comment. As you mention, this is a fairly general introduction to actinic keratoses that does not address the immunological mechanism by which they occur. To correct this, we have completely reformed the introduction and added a text talking about the immunological and molecular mechanisms of actinic keratoses as you can see underlined in green. Furthermore, we have also taken the opportunity to resolve your third comment and have differentiated the concepts of primary and secondary immunomodulation. Thank you again for this great contribution to the review.

Comments 2:  believe Table 1 does not provide relevant information for the article. The description of the sections to be addressed in the review could be included in the material and methodology section.

Response 2: We agree with your comment and have moved this classification table to the materials and methods section. Thank you

Comments 4: An introduction in each section would help understand why the drugs shown are being presented and described.

Response 4: Dear reviewer, we thank you for your comment but we think that we have made an introduction to each section as you can see underlined in yellow. If you think this is insufficient or incorrect, please let us know so we can resolve it.

Comments 5:  The review could improve its quality and readability if the authors include images or diagrams either created by them or obtained from other articles to help exemplify and explain various aspects.

Response 5: Dear reviewer, we agree with you and that is why we have added more tables and figures to the manuscript.

I hope these corrections please you and thank you again for your interest in the manuscript

Kind regards

Reviewer 2 Report

Comments and Suggestions for Authors

The Authors should be commended for a concise, well written and informative paper.

In line 39 the reported incidence of AK is cited at 25%. This may be higher in some cohorts eg organ transplant patients of several years duration or older patients in New Zealand or Australia. 

Author Response

Thank you very much for taking the time to review this manuscript. Please find the  the corresponding revisions below. 

Comments 1: In line 39 the reported incidence of AK is cited at 25%. This may be higher in some cohorts eg organ transplant patients of several years duration or older patients in New Zealand or Australia. 

Response 1: Thank you very much for your comment, indeed the prevalence of actinic keratoses varies in relation to age, location and state of immunosuppression. As you have mentioned, we make it clear in the text that this prevalence can be even higher depending on the court analyzed.

As you can se below:

Actinic keratosis (AK) represents one of the most common premalignant dermatologic conditionsaffecting at around 25% of adult population although this prevalence may be even higher according to some study cohorts attending to immunosuppression grade or ultraviolet exposure (UV) 

Reviewer 3 Report

Comments and Suggestions for Authors

The authors submitted a manuscript describing currently used and possible future topical treatments for actinic keratoses. Given the high prevalence of actinic keratoses and field cancerization, the authors report on a very relevant topic.

The introduction provides sufficient background information on actinic keratoses. The main part of this review describes recent topical treatments and future perspectives. The subsumption appears to be well-structured and provides clear information.

However, I would recommend to add two points:

1.     Line 185-188 regarding pre-treatment of photodynamic therapy: Besides curettage a laser can serve as an ideal means e.g. Erb:YAG or CO2 laser shows even better penetration of 5-ALA afterwards (so called laser-assisted drug delivery), see also Camila de Oliveira Bento et al Photodiagnosis Photodyn Ther 2021, https://doi.org/10.1016/j.pdpdt.2021.102404

2.     Discussion/conclusion part: Since there is growing interest in non-invasive imaging in dermatology,  devices like OCT or LC-OCT facilitate not only a non-invasive diagnosis but also the possibility of monitoring dynamic changes e.g. treatment responses. The implementation of artificial intelligence further promotes the use of this devices (see also Orte Cano, C.; et al. Cancers 2023, https://doi.org/10.3390/ cancers15215264 and Daxenberger et al. Cancers 2023, https://doi.org/10.3390/cancers15184457). I would recommend to include these references.

Author Response

Thank you very much for taking the time to review this manuscript. Please find the detailed responses below and the corresponding revisions 

Comments 1:

 Line 185-188 regarding pre-treatment of photodynamic therapy: Besides curettage a laser can serve as an ideal means e.g. Erb:YAG or CO2 laser shows even better penetration of 5-ALA afterwards (so called laser-assisted drug delivery), see also Camila de Oliveira Bento et al Photodiagnosis Photodyn Ther 2021, https://doi.org/10.1016/j.pdpdt.2021.102404

Response 1: I greatly appreciate this comment since we have not mentioned the other methods to eliminate hyperkeratoses and scabs in actinic keratoses. Currently, the laser has become an increasingly used instrument in dermatology and its usefulness in oncological treatment is increasingly playing a more relevant role. We include the given reference and add the information provided about the laser.

As you can read in 188-193:

Curettage is a common and efficient pre-treatment method. However, there are other methods that can be effective to eliminate hyperkeratosis such as the Erb:YAG or CO2 laser, which in some studies have shown superiority over curettage[31].

Comments 2:    Discussion/conclusion part: Since there is growing interest in non-invasive imaging in dermatology,  devices like OCT or LC-OCT facilitate not only a non-invasive diagnosis but also the possibility of monitoring dynamic changes e.g. treatment responses. The implementation of artificial intelligence further promotes the use of this devices (see also Orte Cano, C.; et al. Cancers 2023, https://doi.org/10.3390/ cancers15215264 and Daxenberger et al. Cancers 2023, https://doi.org/10.3390/cancers15184457). I would recommend to include these references.

Response 2: Thank you very much again for your comment. It is true that we also do not mention the new diagnostic strategies in the cancerization field as well as the artificial intelligence that plays such an important role.
We include a paragraph about these strategies and add the two references.

As you can read in 413- 420:

The diagnosis of AK is eminently clinical, but we cannot ignore that sometimes discerning an AK from Bowen’s disease can be complex if there is a lot of hyperkeratosis or the cancerization field is very damaged. For this reason, there is a general interest in finding non-invasive methods to help the diagnosis and management of these injuries. Devices like OCT or LC-OCT have emerged as tools that allow optimizing efficiency and accuracy in the evaluation of actinic keratosis thanks in turn to the application of artificial intelligence that, through algorithms, allow better categorization of the degree and depth of the lesions, which can help guide treatment and reduce the number of biopsies

Thank you very much for your comments and interest, which have been very helpful.

Round 2

Reviewer 1 Report

Comments and Suggestions for Authors

The authors have adequately answered my questions and used the comments provided to improve the quality of the review. Although I have no further comments on this matter, please review the journal's style for citing images and tables as well as the place where the image captions should appear.